# From Hallucinations to Jailbreaks: Rethinking the Vulnerability of Large Foundation Models

## Abstract

Large foundation models (LFMs) are susceptible to two distinct vulnerabilities: hallucinations and jailbreak attacks. While typically studied in isolation, we observe that defenses targeting one often affect the other, hinting at a deeper connection.

We propose a unified theoretical framework that models jailbreaks as token-level optimization and hallucinations as attention-level optimization. Within this framework, we establish two key propositions: (1) *Similar Loss Convergence*—the loss functions for both vulnerabilities converge similarly when optimizing for target-specific outputs; and (2) *Gradient Consistency in Attention Redistribution*—both exhibit consistent gradient behavior driven by shared attention dynamics.

We validate these propositions empirically on LLaVA-1.5 and MiniGPT-4, showing consistent optimization trends and aligned gradients. Leveraging this connection, we demonstrate that mitigation techniques for hallucinations can reduce jailbreak success rates, and vice versa. Our findings reveal a shared failure mode in LFMs and suggest that robustness strategies should jointly address both vulnerabilities.

## 1 Introduction

Despite their remarkable capabilities, large foundation models (LFMs) remain vulnerable to two distinct yet critical failures: *hallucinations*, where models generate outputs that deviate from factual accuracy (Guan et al., 2024), and *jailbreaks*, where adversaries manipulate model behavior to bypass safety constraints (Shen et al., 2023a). While these vulnerabilities have typically been treated as unrelated problems—one stemming from internal misalignment and the other from external manipulation—they both reflect a fundamental loss of control in model behavior.

Surprisingly, we observe that these two failures are not only interrelated but also influence each other in practice. As shown in Fig. 1, applying a jailbreak defense prompt (Xie et al., 2023) can mitigate hallucinations in visual reasoning tasks, allowing models like LLaVA-1.5 (Liu et al., 2024b) and MiniGPT-4 (Zhu et al., 2023a) to correctly judge visual illusions. Conversely, incorporating a hallucination mitigation strategy (Huang et al., 2024) can cause the model to reject previously successful jailbreak prompts. These bidirectional effects suggest a shared underlying mechanism, challenging the conventional view that hallucinations and jailbreaks arise from independent causes.

Prior research has extensively analyzed hallucinations through attention drift (Chuang et al., 2024; Yuksekgonul et al., 2023), internal uncertainty (Azaria & Mitchell, 2023; He et al., 2024), and token-level inconsistencies (Quevedo et al., 2024), proposing mitigation strategies accordingly. Separately, jailbreak attacks have been explored in both white-box (Zou et al., 2023; Zhu et al., 2023b) and black-box (Jin et al., 2024a;c; Chao et al., 2023) settings, with defenses ranging from prompt filtering (Xie et al., 2023) to obfuscation detection (Jin et al., 2024b). However, no prior work systematically investigates the interplay between these two vulnerabilities or the possibility that they arise from a shared optimization structure.

In this paper, we present a unified theoretical framework that formalizes this connection. We model jailbreaks as token-level optimization problems that manipulate output distributions, and hallucinations as attention-level optimization problems that perturb internal focus. This framework leads to two key theoretical propositions: **(1) Similar Loss Convergence**—the loss functions of

Figure 1: Example of the interplay between hallucinations and jailbreaks. (a) LLaVA-1.5 and MiniGPT-4 misjudge the lengths of the yellow lines due to hallucinations, which can be corrected using a jailbreak defense prompt. (b) Carefully crafted jailbreak prompts (orange-colored) bypass safeguards to produce harmful outputs, while hallucination mitigation neutralizes these prompts, resulting in refusal responses.

hallucinations and jailbreaks converge under similar optimization dynamics, suggesting that both are intrinsic to LFM behavior; and **(2) Gradient Consistency in Attention Redistribution**—the gradients guiding both phenomena exhibit consistent patterns, implying that perturbations affecting one are likely to influence the other.

We empirically validate these propositions on two open-source vision-language models, LLaVA-1.5 and MiniGPT-4. Our experiments reveal that mitigation strategies designed for hallucinations can reduce the success rate of jailbreak attacks, and vice versa. This cross-effect generalizes across architectures and attack variants, reinforcing the validity of our framework. These findings open a new perspective for understanding model vulnerabilities and suggest that robustness interventions should consider hallucinations and jailbreaks jointly.

**Our key contributions are as follows:**

- **Exploration of the Hallucination-Jailbreak Interplay:** We are the first to identify and formalize the optimization-based connection between hallucinations and jailbreaks, challenging the conventional view that treats them as distinct.
- **Unified Theoretical Framework:** We introduce a modeling framework that casts jailbreaks as token-level optimization and hallucinations as attention-level optimization, revealing their shared structure.
- **Theoretical and Empirical Validation:** We establish two theoretical propositions characterizing convergence and gradient behavior, and empirically validate them across two representative LFMs.
- **Cross-Domain Mitigation Insights:** We demonstrate that mitigating one vulnerability can enhance robustness against the other, offering practical guidance for safer foundation model deployment.

## 2 RELATED WORK

**Jailbreak Attacks**. Jailbreak attacks can be classified into manual and automated methods. Early jailbreaks involved manually refining prompts through trial-and-error, exploiting randomness over multiple attempts (Li et al., 2023; Shen et al., 2023b), with empirical studies quantifying their effectiveness (Wei et al., 2023; Shen et al., 2023a). Automated jailbreaks advance this process: Zou et al. Zou et al. (2023) proposed gradient-based attacks optimizing token positions, while Chao et al. Chao et al. (2023) refined prompts iteratively using prior responses. Other methods focus on role-playing strategies (Jin et al., 2024a) or adversarial querying in black-box settings (Hayase et al., 2024). Recent efforts leverage cryptographic techniques to obfuscate malicious prompts, improving evasion against detection models (Ren et al., 2024; Li et al., 2024; Yuan et al., 2023; Handa et al., 2024; Jin et al., 2024c). Vision-language models are similarly susceptible to adversarial image perturbations that induce harmful outputs (Carlini et al., 2024; Zhao et al., 2023; Qi et al., 2023; Schlarmann & Hein, 2023). Specialized benchmarks facilitate systematic jailbreak evaluations (Luo et al., 2024).

**Hallucination**. Hallucination analysis spans multiple perspectives, including attention, internal activations, and token-level features. At the attention level, Chuang et al. Chuang et al. (2024) introduced Lookback Lens, quantifying attention shifts between input context and generated tokens, while Yuksekgonul et al. Yuksekgonul et al. (2023) modeled factuality detection using Constraint Satisfaction Problems (CSPs). Internal state-based approaches examine hidden activations to infer model certainty (Azaria & Mitchell, 2023; Beigi et al., 2024; He et al., 2024; Duan et al., 2024). Methods such as ACT (Wang et al., 2024a) and INSIDE (Chen et al., 2024) introduce activation steering and eigenvalue-based feature clipping to improve factuality. Token-level analyses employ probability-based indicators to detect hallucinations efficiently (Quevedo et al., 2024; Su et al., 2024), emphasizing statistical consistency over semantic interpretation.

## 3 PRELIMINARIES

In this section, we formalize hallucinations and jailbreaks within a shared optimization framework over large foundation models (LFMs). While these vulnerabilities manifest differently—hallucinations as factual drift and jailbreaks as adversarial instruction bypass—they both result from perturbations that misalign model outputs with intended behavior. Our goal is to define structured loss functions for each vulnerability type, setting the stage for the theoretical analysis of their convergence and optimization dynamics in Section 4.

### 3.1 JAILBREAK ATTACK

We model jailbreak attacks on both large language models (LLMs) and vision-language models (VLMs) using a unified input representation. Let $\mathbf{x}^{\mathbf{t}}$ denote the textual input, which is encoded into textual embeddings $\mathbf{H}^{\mathbf{t}}_{1:n}$. If the model is multi-modal, visual input $\mathbf{x}^{\mathbf{v}}$ is processed through a feature extractor $g$ and projected into the language space to produce visual embeddings $\mathbf{H}^{\mathbf{v}}_{1:l} = \mathbf{W} \cdot g(\mathbf{x}^{\mathbf{v}})$. The complete input is then represented as $[\mathbf{H}^{\mathbf{v}}_{1:l}, \mathbf{H}^{\mathbf{t}}_{1:n}]$; for unimodal LLMs, this reduces to $[\varnothing, \mathbf{H}^{\mathbf{t}}_{1:n}]$.

The model generates an output sequence $\mathbf{y}$ with probability:

$$p(\mathbf{y} \mid [\mathbf{H}^{\mathbf{v}}_{1:l}, \mathbf{H}^{\mathbf{t}}_{1:n}]) = \prod_{i=1} p(\mathbf{y}_i \mid \mathbf{y}_{1:i-1}, [\mathbf{H}^{\mathbf{v}}_{1:l}, \mathbf{H}^{\mathbf{t}}_{1:n}]) \tag{1}$$

In aligned LFMs, output behavior is implicitly shaped by a latent reward model $\mathcal{R}^*$, which reflects alignment with human preferences. Jailbreak attacks aim to induce harmful outputs by perturbing a benign input $[\hat{\mathbf{H}}^{\mathbf{v}}_{1:l}, \hat{\mathbf{H}}^{\mathbf{t}}_{1:n}]$ into an adversarial input $[\tilde{\mathbf{H}}^{\mathbf{v}}_{1:l}, \tilde{\mathbf{H}}^{\mathbf{t}}_{1:n}]$ such that the model produces low-reward or harmful outputs:

$$\mathbf{y}^{\star} = \min \mathcal{R}^*(\mathbf{y} \mid [\tilde{\mathbf{H}}^{\mathbf{v}}_{1:l}, \tilde{\mathbf{H}}^{\mathbf{t}}_{1:n}]) \tag{2}$$

Instead of accessing $\mathcal{R}^*$ directly, we optimize over the likelihood of the adversarial output $\mathbf{y}^{\star}$, defining the jailbreak loss as:

$$\mathcal{L}^{adv}([\tilde{\mathbf{H}}^{\mathbf{v}}_{1:l}, \tilde{\mathbf{H}}^{\mathbf{t}}_{1:n}]) = -\log p(\mathbf{y}^{\star} \mid [\tilde{\mathbf{H}}^{\mathbf{v}}_{1:l}, \tilde{\mathbf{H}}^{\mathbf{t}}_{1:n}]) \tag{3}$$

The optimal adversarial embedding is obtained by minimizing this objective over a constrained perturbation set $\mathcal{A}$:

$$[\tilde{\mathbf{H}}^{\mathbf{v}}_{1:k}, \tilde{\mathbf{H}}^{\mathbf{t}}_{1:n}] = \underset{[\tilde{\mathbf{H}}^{\mathbf{v}}_{1:k}, \tilde{\mathbf{H}}^{\mathbf{t}}_{1:n}] \in \mathcal{A}([\hat{\mathbf{H}}^{\mathbf{v}}_{1:k}, \hat{\mathbf{H}}^{\mathbf{t}}_{1:n}])}{\arg\min} \mathcal{L}^{adv}([\tilde{\mathbf{H}}^{\mathbf{v}}_{1:k}, \tilde{\mathbf{H}}^{\mathbf{t}}_{1:n}]) \tag{4}$$

This formulation captures the adversary's goal: to shift the model's output distribution toward unsafe completions using perturbations at the embedding level, which is a common setting in both white-box and black-box jailbreak literature (Zou et al., 2023; Zhu et al., 2023b; Jin et al., 2024a).

### 3.2 HALLUCINATION

We model hallucinations as attention-level perturbations that misallocate focus across the input sequence. Consider the input embedding $[\mathbf{H}^{\mathbf{v}}_{1:l}, \mathbf{H}^{\mathbf{t}}_{1:n}]$ in a $d_e$-dimensional space. For simplicity, we

analyze a single attention block (Yao et al., 2023), where attention weights $A_{ij}$ and output embedding $o_i$ are computed as:

$$A_{ij} = \frac{\exp\left((W_Q\mathbf{H}_i)^{\mathrm{T}}(W_K\mathbf{H}_j)\right)}{\sum_{k=1}^{n+l} \exp\left((W_Q\mathbf{H}_i)^{\mathrm{T}}(W_K\mathbf{H}_k)\right)} \tag{5}$$

$$o_i = \sum_{j=1}^{n+l} A_{ij} \cdot (W_V\mathbf{H}_j) \tag{6}$$

where $W_Q$, $W_K$, and $W_V$ are projection matrices. Hallucinations occur when attention is misallocated, reducing focus on relevant regions and increasing focus on semantically unrelated tokens.

To simulate this behavior, we introduce perturbation vectors $\Delta^{\mathbf{t}}$ and $\Delta^{\mathbf{v}}$ to the input embeddings:

$$\tilde{\mathbf{H}}^{\mathbf{t}}_{1:n} = \mathbf{H}^{\mathbf{t}}_{1:n} + \Delta^{\mathbf{t}}, \quad \tilde{\mathbf{H}}^{\mathbf{v}}_{1:k} = \mathbf{H}^{\mathbf{v}}_{1:l} + \Delta^{\mathbf{v}} \tag{7}$$

The perturbed embeddings $[\tilde{\mathbf{H}}^{\mathbf{v}}_{1:l}, \tilde{\mathbf{H}}^{\mathbf{t}}_{1:n}]$ result in updated attention scores:

$$A^{\Delta}_{ij} = \frac{\exp\left((W_Q\tilde{\mathbf{H}}_i)^{\mathrm{T}}(W_K\mathbf{H}_j)\right)}{\sum_{k=1}^{n+l} \exp\left((W_Q\tilde{\mathbf{H}}_i)^{\mathrm{T}}(W_K\mathbf{H}_k)\right)} \tag{8}$$

and corresponding output embeddings $\tilde{o}_i = \sum_{j=1}^{n+l} A^{\Delta}_{ij} \cdot (W_V\mathbf{H}_j)$.

We define a hallucination loss to guide attention toward a target position $t$ while minimizing attention to non-targets:

$$\mathcal{L}^{hallu} = \sum_{i=1}^{m} \left( -\log\left(A^{\Delta}_{it}\right) + \lambda \sum_{j\neq t} \log\left(A^{\Delta}_{ij}\right) \right) \tag{9}$$

where $m$ is the number of output tokens, and $\lambda$ balances focus versus suppression. This loss operationalizes the notion that hallucinations can be modeled and mitigated by enforcing structured attention reallocation.

In the next section, we analyze the interplay between these loss functions, showing that hallucinations and jailbreaks, though distinct in manifestation, share similar optimization dynamics and gradient behaviors under embedding perturbations.

## 4 INTERPLAY BETWEEN JAILBREAK AND HALLUCINATION

With the loss functions $\mathcal{L}^{adv}$ (Eq. 3) and $\mathcal{L}^{hallu}$ (Eq. 9) formalized, we now analyze their underlying optimization behavior. Our goal is to determine whether hallucinations and jailbreaks—despite arising from different components of the model—exhibit structurally similar dynamics under perturbation. We approach this question by studying both the convergence of their loss functions and the alignment of their gradients. If both loss functions converge in similar regimes and share directional gradient components, this would support the hypothesis that hallucinations and jailbreaks reflect a common vulnerability intrinsic to LFMs.

### 4.1 LOSS CONVERGENCE ANALYSIS

Recall that output token probabilities in LFMs are determined via the softmax over decoder outputs (Eq. 1). When perturbations alter the attention-derived hidden states $o_i$, the softmax logits $\tilde{o}_i$ change accordingly. Expanding the jailbreak loss $\mathcal{L}^{adv}$ (Eq. 3) for a given target sequence $\mathbf{y}^*$, we obtain:

$$\mathcal{L}^{adv}([\tilde{\mathbf{H}}^{\mathbf{v}}_{1:l}, \tilde{\mathbf{H}}^{\mathbf{t}}_{1:n}]) = -\sum_{i=1}^{m} \left( [W_{\text{out}}\tilde{o}_i]_{\mathbf{y}^*_i} - \log \sum_{j=1}^{|V|} \exp\left([W_{\text{out}}\tilde{o}_i]_j\right) \right)$$

where $[W_{\text{out}}\tilde{o}_i]_{\mathbf{y}^*_i}$ is the logit corresponding to the target token $\mathbf{y}^*_i$. This formulation reveals that the jailbreak loss penalizes deviations of model outputs from a harmful target sequence.

In contrast, the hallucination loss $\mathcal{L}^{hallu}$ modulates attention weights to increase focus on the target position $t$ while reducing focus elsewhere. Both objectives thus impose structure: one over token likelihoods, the other over attention distributions. To analyze their convergence relationship, we consider the following stylized setting.

**Proposition 4.1** (Similar Loss Convergence). *Assume the following proportional scaling conditions hold:*

$$[W_{out}\tilde{o}_i]_j = \beta_j [W_{out}\tilde{o}_i]_{\mathbf{y}_i^*}, \quad \forall j \neq \mathbf{y}_i^* \tag{10}$$

$$A_{ij}^\Delta = \eta_j A_{it}^\Delta, \quad \forall j \neq t \tag{11}$$

*where $\beta_j, \eta_j \in [0, 1)$ are dynamic scaling factors. Suppose further that:*

- *The non-target logits decay: $\sum_{j \neq \mathbf{y}_i^*} \beta_j \to 0$;*
- *The non-target attention decays: $\sum_{j \neq t} \eta_j \to 0$;*
- *The regularization term satisfies $\lambda \to 0$, with $\lambda \ll \beta_j$ for all $j$.*

*Then:*

$$\lim_{\sum_{j \neq t} \eta_j \to 0} \mathcal{L}^{hallu} = \lim_{\sum_{j \neq \mathbf{y}_i^*} \beta_j \to 0} \mathcal{L}^{adv} \tag{12}$$

*Justification and Interpretation.* These assumptions capture common behaviors observed during high-confidence decoding in LFMs. In both autoregressive generation and alignment-finetuned models, output logits for the most probable token often dominate the softmax distribution—this is reflected in the assumption that all other logits are proportional to the dominant one via diminishing factors $\beta_j$. This behavior becomes especially pronounced under adversarial optimization (e.g., jailbreak attacks), where the optimization objective amplifies this token-level concentration.

Similarly, in transformer attention, empirical studies have shown that attention distributions tend to become increasingly peaked around salient inputs during optimization (Michel et al., 2019; Clark et al., 2019), particularly in late-stage fine-tuned or instruction-following models. Our assumption that non-target attention scores scale with $\eta_j \to 0$ mirrors this behavior, modeling the vanishing influence of irrelevant context as attention concentrates on a single dominant position.

Together, these assumptions idealize a regime of softmax sparsification—a known emergent property in highly confident LFMs—where both attention and decoding behavior converge toward target-specific outputs. While stylized, these assumptions are consistent with trends observed across many autoregressive decoding trajectories and adversarial attacks.

Under this setting, both losses reduce to negative log-attention or logit terms at target positions. This proposition establishes that *hallucination and jailbreak objectives converge to similar scalar functions*, indicating that both reflect a fundamental optimization tendency of LFMs rather than isolated bugs.

## 4.2 GRADIENT ALIGNMENT

Beyond convergence, we analyze whether both losses respond similarly to perturbations. Let us examine the gradients of $\mathcal{L}^{adv}$ and $\mathcal{L}^{hallu}$ with respect to input embeddings, specifically through their impact on attention.

**Proposition 4.2** (Gradient Consistency in Attention Redistribution). *Let the shared component of attention-driven gradient response be:*

$$\Delta_{ij} = A_{ij}^\Delta \left( W_Q^T W_K \mathbf{H}_j - \sum_{k=1}^{n+l} A_{ik}^\Delta W_Q^T W_K \mathbf{H}_k \right) \cdot W_V \mathbf{H}_j \tag{13}$$

*Then, for sufficiently large $\lambda$, the dominant gradient contribution in $\nabla\mathcal{L}^{hallu}$ aligns with that in $\nabla\mathcal{L}^{adv}$, reinforcing their shared sensitivity to attention perturbation.*

*Interpretation.* $\Delta_{ij}$ describes the influence of a given input token $j$ on the output $i$ via attention. It captures how shifting attention toward or away from certain tokens affects the overall embedding update. The fact that this term appears in both losses implies that *their optimization directions are*

*aligned.* When perturbing inputs (either to induce hallucinations or enable jailbreaks), both losses push the model in similar directions in embedding space.

Together, Propositions 4.1 and 4.2 demonstrate that hallucinations and jailbreaks—despite targeting different vulnerabilities—share optimization dynamics and respond similarly to embedding perturbations. This finding provides theoretical justification for the empirical observation that mitigation of one vulnerability can influence the other. Detailed derivations and proofs are provided in Appendix B and Appendix C.

## 5 EXPERIMENTS

### 5.1 EXPERIMENTAL SETUP

**Models.** We conduct experiments on two open-source vision-language models: LLaVA-1.5 (`llava-1.5-7b-hf`) (Liu et al., 2024a) and MiniGPT-4 (`minigpt4-vicuna-7B`) (Zhu et al., 2023a). These models are well-suited to our analysis due to their open-access architecture and support for gradient-based manipulation. This allows us to directly optimize attention distributions and token-level outputs—key components of our theoretical framework in Proposition 4.1 and Proposition 4.2.

**Dataset.** To jointly study hallucination and jailbreak behavior, we use 50 commonsense reasoning prompts from (Yao et al., 2023). These prompts are selected to induce semantically grounded reasoning, where both hallucinations (due to ambiguous or misaligned attention) and jailbreaks (via adversarial suffixes) can be systematically triggered. This shared prompt base enables side-by-side evaluation of both loss behaviors under controlled conditions.

**Implementation Details.** For hallucination optimization, we maximize the attention redistribution loss $\mathcal{L}^{hallu}$ (Eq. 9) by defining a fixed attention target position. For jailbreaks, we implement the GCG method (Zou et al., 2023), which iteratively appends a trainable suffix to the prompt to increase the likelihood of a predefined adversarial target output. The suffix is initialized as a sequence of 20 exclamation marks. To isolate the optimization behavior of text modalities and ensure fair comparison across settings, we disable visual inputs during all gradient-based experiments.

### 5.2 EXPERIMENTAL ANALYSIS OF SIMILAR LOSS CONVERGENCE

To empirically validate Proposition 4.1, which posits that hallucination and jailbreak losses converge under similar optimization dynamics, we conduct experiments on 50 commonsense reasoning questions. Each question is used to guide gradient-based optimization over 80 steps toward a predefined target output for both hallucination and jailbreak objectives.

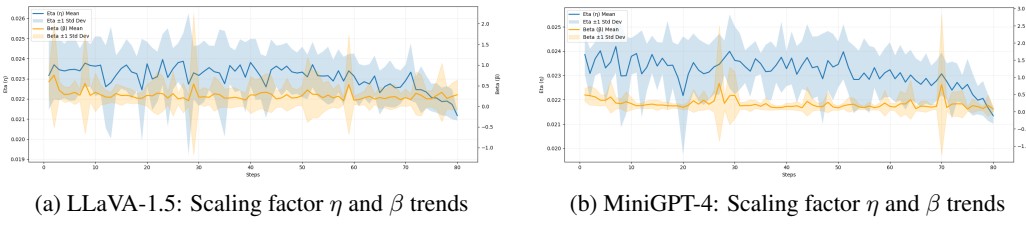

(a) LLaVA-1.5: Scaling factor $\eta$ and $\beta$ trends      (b) MiniGPT-4: Scaling factor $\eta$ and $\beta$ trends

Figure 2: Scaling factors $\eta$ and $\beta$ over 80 opt. steps for LLaVA-1.5 and MiniGPT-4.

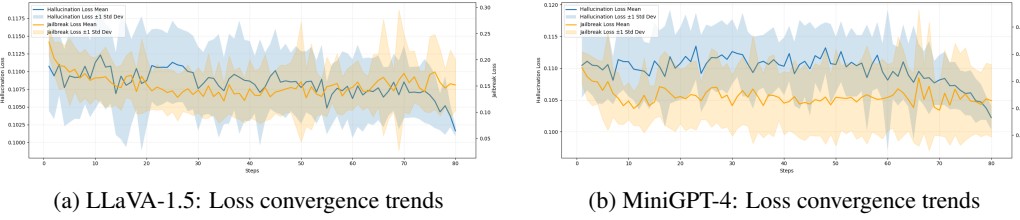

(a) LLaVA-1.5: Loss convergence trends      (b) MiniGPT-4: Loss convergence trends

Figure 3: Hallucination and jailbreak losses over 80 opt. steps for LLaVA-1.5 and MiniGPT-4.

We evaluate this convergence using two key metrics:

- **Scaling Factors $\eta$ and $\beta$:** These quantify the concentration dynamics of attention and output logits, respectively. Specifically, we define $\eta = \sum_{j \neq t} A_{ij}^{\Delta}/A_{it}^{\Delta}$ as the ratio of residual attention mass distributed to non-target tokens, and $\beta = \sum_{j \neq \mathbf{y}_i^*} \text{logit}_j / \text{logit}_{\mathbf{y}_i^*}$ as the corresponding ratio for the output logits. Both quantities reflect the extent to which the model's focus shifts toward the intended target across optimization steps. Smaller values of $\eta$ and $\beta$ indicate stronger concentration on the correct token in both attention and prediction layers.
- **Loss Convergence:** We track the hallucination loss $\mathcal{L}^{\text{hallu}}$ and the jailbreak loss $\mathcal{L}^{\text{adv}}$ over gradient descent iterations to determine whether the model is minimizing both losses simultaneously. Convergent behavior would suggest that the same optimization process is capable of reducing both vulnerabilities.

**Scaling behavior of $\eta$ and $\beta$.** As shown in Fig. 2, both LLaVA-1.5 and MiniGPT-4 exhibit consistent decay patterns in $\eta$ and $\beta$ throughout the optimization process. For LLaVA-1.5 (Fig. 2a), the attention scaling factor $\eta$ initially remains stable, then declines steadily toward zero, indicating a progressive concentration of attention on the target token. The logit scaling factor $\beta$ also decreases over time, although with slightly greater variance, reflecting the dynamic nature of output prediction adjustments. A similar trend is observed for MiniGPT-4 (Fig. 2b), where both metrics decrease smoothly, suggesting that attention and output spaces become increasingly aligned with the target across training iterations.

**Convergence of hallucination and jailbreak losses.** The loss curves in Fig. 3 further reinforce this finding. For both LLaVA-1.5 (Fig. 3a) and MiniGPT-4 (Fig. 3b), the hallucination loss $\mathcal{L}^{hallu}$ and jailbreak loss $\mathcal{L}^{\text{adv}}$ exhibit synchronous and monotonic declines. Despite potential differences in initial loss magnitudes and convergence rates across models, both losses decrease steadily, suggesting that a shared optimization trajectory underlies both phenomena. This coupled convergence supports the hypothesis that the vulnerabilities are structurally linked rather than independent.

**Interpretation.** Together, these empirical results offer strong support for Proposition 4.1. The observed reductions in $\eta$ and $\beta$ confirm that non-target influence is suppressed in both attention and output layers as training progresses. Meanwhile, the tandem decline of $\mathcal{L}^{hallu}$ and $\mathcal{L}^{adv}$ across models and settings indicates that hallucinations and jailbreak behaviors are not isolated quirks but rather co-emerge from a shared optimization mechanism in large foundation models. This alignment between theory and experiment strengthens the view that targeted manipulations in one modality (e.g., attention) can influence vulnerabilities in another (e.g., output generation), revealing a deeper coupling between internal representations and emergent behavior in these models.

## 5.3 EXPERIMENTAL ANALYSIS OF GRADIENT CONSISTENCY IN ATTENTION REDISTRIBUTION

To evaluate Proposition 4.2, which posits that hallucination and jailbreak losses share aligned gradient directions under perturbation, we analyze optimization behavior under varying regularization strengths $\lambda \in \{0.1, 1, 10, 100\}$. Specifically, we examine: (1) The loss trajectories of $\mathcal{L}^{hallu}$ and $\mathcal{L}^{adv}$ over 80 optimization steps; (2) The alignment between their gradients using cosine similarity and Spearman correlation.

**Loss trends under varying $\lambda$.** Fig. 4 shows that both losses decrease consistently throughout optimization. For LLaVA-1.5 (Fig. 4a), hallucination loss exhibits initial fluctuations but converges steadily, while jailbreak loss follows a parallel descent. MiniGPT-4 (Fig. 4b) shows a similar pattern. Notably, as $\lambda$ increases—placing greater emphasis on target-vs-nontarget contrast in $\mathcal{L}^{hallu}$—the alignment of the two loss curves becomes more pronounced. This supports the theoretical prediction that stronger regularization enhances coupling between the two objectives.

**Gradient similarity.** Table 1 reports cosine similarity and Spearman rank correlation between the gradients of hallucination loss $\mathcal{L}^{hallu}$ and jailbreak loss $\mathcal{L}^{adv}$. Cosine similarity captures the directional alignment of gradients in parameter space—i.e., whether both losses suggest updates in similar directions. Spearman correlation measures the consistency in the rank order of gradient magnitudes, reflecting whether both losses emphasize updates to the same parameters. Across all tested values of $\lambda$, both similarity metrics remain consistently high (above 0.93), indicating strong agreement in both the direction and structure of gradient updates. This suggests that minimizing either

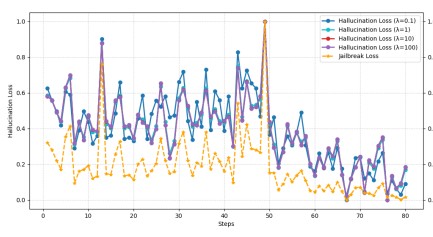 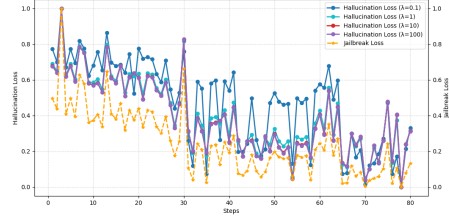

(a) LLaVA-1.5: Loss trajectories          (b) MiniGPT-4: Loss trajectories

Figure 4: Hallucination and jailbreak losses over 80 optimization steps under different values of $\lambda$.

loss leads to similar movement in the model's parameter space. As $\lambda$ increases, the alignment becomes even more pronounced, implying that the joint optimization trajectory becomes increasingly unified under stronger interpolation. These trends offer empirical support for Proposition 4.2, confirming that hallucination and jailbreak behaviors are not driven by separate gradients but instead co-emerge from shared optimization dynamics. This reinforces the view that both vulnerabilities are manifestations of the same underlying attention-weighted representation shift, shaped by a unified set of gradients.

**Interpretation.** These findings provide empirical validation for Proposition 4.2, extending beyond convergence trends to reveal gradient-level alignment between hallucination and jailbreak losses. This suggests that such failure modes are not isolated

Table 1: Cosine similarity and Spearman correlation of hallucination and jailbreak gradients across $\lambda$.

| Models | Cosine Similarity | | | | Spearman Correlation | | | |
|---|---|---|---|---|---|---|---|---|
| | 0.1 | 1 | 10 | 100 | 0.1 | 1 | 10 | 100 |
| LLaVA-1.5 | 0.957 | 0.957 | 0.962 | 0.983 | 0.934 | 0.935 | 0.957 | 0.966 |
| MiniGPT-4 | 0.934 | 0.934 | 0.939 | 0.968 | 0.958 | 0.958 | 0.973 | 0.977 |

artifacts but arise from a shared optimization geometry—specifically, coordinated shifts in attention-weighted representations. Recognizing this structure opens the door to unified mitigation strategies that target root causes rather than symptoms.

Table 2: Correctness Improvements on HallusionBench and AutoHallusion Benchmarks

| Models | Benchmarks | Initial Correct. | Defense/Mitigation | Post Correct. ($\uparrow$) |
|---|---|---|---|---|
| LLaVA-1.5 | HallusionBench | 19.6% | OPERA | 42.8% (23.2%$\uparrow$) |
| | | | VCD | 35.6% (16.0%$\uparrow$) |
| | | | Goal Prioritization | 32.4% (12.8%$\uparrow$) |
| | | | AdaShield-Static | 30.2% (10.6%$\uparrow$) |
| | AutoHallusion | 15.4% | OPERA | 40.0% (24.6%$\uparrow$) |
| | | | VCD | 32.4% (17.0%$\uparrow$) |
| | | | Goal Prioritization | 28.6% (13.2%$\uparrow$) |
| | | | AdaShield-Static | 26.8% (11.4%$\uparrow$) |
| MiniGPT-4 | HallusionBench | 15.2% | OPERA | 41.2% (26.0%$\uparrow$) |
| | | | VCD | 33.2% (18.0%$\uparrow$) |
| | | | Goal Prioritization | 30.4% (15.2%$\uparrow$) |
| | | | AdaShield-Static | 28.6% (13.4%$\uparrow$) |
| | AutoHallusion | 12.8% | OPERA | 37.8% (25.0%$\uparrow$) |
| | | | VCD | 36.4% (23.6%$\uparrow$) |
| | | | Goal Prioritization | 27.6% (14.8%$\uparrow$) |
| | | | AdaShield-Static | 25.8% (13.0%$\uparrow$) |

## 5.4 CROSS-PHENOMENON MITIGATION: GENERALIZING DEFENSES BETWEEN HALLUCINATIONS AND JAILBREAKS

Our theoretical analysis suggests that hallucinations and jailbreaks share optimization structures and gradient behavior. A key implication is that mitigation strategies developed for one vulnerability may generalize to the other. We now validate this hypothesis empirically in both directions. These cross-domain evaluations provide strong functional support for our theoretical framework.

### 5.4.1 MITIGATING HALLUCINATIONS VIA JAILBREAK DEFENSES

We examine whether jailbreak defenses-designed to resist adversarial prompt injections—can also reduce hallucination in VLMs. We evaluate performance on HallusionBench (Guan et al., 2024) and AutoHallusion (Wu et al., 2024), two curated datasets containing factual QA tasks where models frequently produce semantically inconsistent or misleading responses.

Table 3: Effectiveness of Defense and Mitigation Methods in Reducing Jailbreak Success Rates

| Models | Attack | Initial ASR | Defense/Mitigation | Post ASR (↓) |
|---|---|---|---|---|
| LLaVA-1.5 | FigStep | 82.20% | OPERA | 34.8% (47.4%↓) |
| | | | VCD | 30.6% (51.6%↓) |
| | | | Goal Prioritization | 36.4% (45.8%↓) |
| | | | AdaShield-Static | 27.8% (54.4%↓) |
| | MML | 86.4% | OPERA | 40.4% (46.0%↓) |
| | | | VCD | 37.6% (48.8%↓) |
| | | | Goal Prioritization | 43.0% (43.4%↓) |
| | | | AdaShield-Static | 37.2% (49.2%↓) |
| MiniGPT-4 | FigStep | 74.8% | OPERA | 27.4% (47.4%↓) |
| | | | VCD | 25.2% (49.6%↓) |
| | | | Goal Prioritization | 28.6% (46.2%↓) |
| | | | AdaShield-Static | 23.6% (51.2%↓) |
| | MML | 85.4% | OPERA | 45.0% (40.4%↓) |
| | | | VCD | 41.8% (43.6%↓) |
| | | | Goal Prioritization | 48.4% (37.0%↓) |
| | | | AdaShield-Static | 31.6% (53.8%↓) |

We measure hallucination severity using the **Correctness** metric, defined as the percentage of outputs that align with ground-truth semantics. Each benchmark contains 500 QA items. For each model and method, we report both the *Initial Correctness* (before any defense is applied) and the *Post Correctness*, along with the improvement (↑) as a percentage point increase.

We compare two jailbreak-specific defenses—Goal Prioritization (Xie et al., 2023) and AdaShield-Static (Wang et al., 2024b)—with two hallucination-specific mitigation strategies: OPERA (Huang et al., 2024) and VCD (Leng et al., 2024), both based on attention reallocation. Prompt templates for jailbreak defenses are listed in Appendix E.

**Findings.** Table 2 shows that jailbreak defenses also mitigate hallucinations: on HallusionBench with LLaVA-1.5, Goal Prioritization and AdaShield-Static improve correctness by 12.8% and 10.6%, while attention-focused OPERA achieves 23.2%. This supports the claim that output- and attention-level failures are structurally coupled.

### 5.4.2 DEFENDING JAILBREAKS VIA HALLUCINATION MITIGATION

We next evaluate whether hallucination mitigation methods can also reduce susceptibility to jailbreak attacks. Experiments are conducted on SafeBench (Xu et al., 2022), a standard benchmark of adversarial prompts designed to elicit harmful responses from VLMs. We report performance in terms of **Attack Success Rate (ASR)**: the percentage of harmful prompts that yield unsafe model completions. We compare *Initial ASR* with *Post ASR* and report relative reduction (↓). Models are evaluated under two strong attack types—FigStep (Gong et al., 2023) and MML (Wang et al., 2024c)—with the same four defenses: hallucination mitigators (OPERA, VCD) and jailbreak defenses (Goal Prioritization, AdaShield-Static).

**Findings.** Table 3 shows that hallucination mitigation methods achieve substantial reductions in ASR—often rivaling or outperforming jailbreak-specific defenses. For example, under FigStep on MiniGPT-4, VCD reduces ASR from 74.8% to 25.2% (49.6% ↓), nearly matching AdaShield-Static (51.2% ↓). These results validate Proposition 4.2, showing that attention reallocation not only corrects factual drift but also impedes adversarial goal injection.

Additional results on larger models and alternative jailbreak types are provided in Appendix D.

## 6 CONCLUSION

In this paper, we investigate the interplay between hallucinations and jailbreak attacks in LFMs, revealing their intrinsic connection through shared optimization dynamics. Modeling jailbreaks as token-level optimization and hallucinations as attention-level optimization, we establish a unified framework with two propositions: similar loss convergence and gradient consistency in attention redistribution. We validate these through theoretical analysis and experiments on LLaVA-1.5 and MiniGPT-4, showing that certain jailbreak defenses also mitigate hallucinations. This challenges the conventional view that treats them separately, emphasizing the need for a holistic approach to improving robustness.

ETHICS STATEMENT

This work adheres to the ICLR Code of Ethics. In this study, no human subjects or animal experimentation were involved. All datasets used were sourced in compliance with relevant usage guidelines, ensuring no violation of privacy. We have taken care to avoid any biases or discriminatory outcomes in our research process. No personally identifiable information was used, and no experiments were conducted that could raise privacy or security concerns. We are committed to maintaining transparency and integrity throughout the research process.

REPRODUCIBILITY STATEMENT

We have made every effort to ensure that the results presented in this paper are reproducible. All code and datasets have been made publicly available in an anonymous repository to facilitate replication and verification. The experimental setup, including training steps, model configurations, and hardware details, is described in detail in the paper. We have also provided a full description of our proposed method to assist others in reproducing our experiments.

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

## A    LLM USAGE

Large Language Models (LLMs) were used to aid in the writing and polishing of the manuscript. Specifically, we used an LLM to assist in refining the language, improving readability, and ensuring clarity in various sections of the paper. The model helped with tasks such as sentence rephrasing, grammar checking, and enhancing the overall flow of the text.

It is important to note that the LLM was not involved in the ideation, research methodology, or experimental design. All research concepts, ideas, and analyses were developed and conducted by the authors. The contributions of the LLM were solely focused on improving the linguistic quality of the paper, with no involvement in the scientific content or data analysis.

The authors take full responsibility for the content of the manuscript, including any text generated or polished by the LLM. We have ensured that the LLM-generated text adheres to ethical guidelines and does not contribute to plagiarism or scientific misconduct.

## B    DETAILED DERIVATIONS

### B.1    TAYLOR SERIES EXPANSION OF HALLUCINATION LOSS AND JAILBREAK LOSS

According to Eq. 27, when $\sum_{j\neq \mathbf{y}_i^*}^{|V|} \beta_j \to 0$, we can get the Taylor Series expansion. We consider the term:

$$\sum_{j\neq \mathbf{y}_i^*}^{|V|} \exp(\beta_j [W_{\text{out}}\tilde{o}_i]_{\mathbf{y}_i^*}) \tag{14}$$

Expanding the exponential function using its Taylor series:

$$\exp(\beta_j [W_{\text{out}}\tilde{o}_i]_{\mathbf{y}_i^*}) = 1 + \beta_j [W_{\text{out}}\tilde{o}_i]_{\mathbf{y}_i^*} + \frac{1}{2}\beta_j^2 [W_{\text{out}}\tilde{o}_i]_{\mathbf{y}_i^*}^2 + O(\beta_j^3) \tag{15}$$

Substituting this into Eq. 27, we obtain:

$$\mathcal{L}^{adv} = \sum_{i=1}^{m}\left(-[W_{\text{out}}\tilde{o}_i]_{\mathbf{y}_i^*} + \log\sum_{j\neq \mathbf{y}_i^*}^{|V|}\left(1 + \beta_j [W_{\text{out}}\tilde{o}_i]_{\mathbf{y}_i^*} + \frac{1}{2}\beta_j^2 [W_{\text{out}}\tilde{o}_i]_{\mathbf{y}_i^*}^2 + O(\beta_j^3)\right)\right) \tag{16}$$

### B.2    TAYLOR SERIES EXPANSION OF HALLUCINATION LOSS AND JAILBREAK LOSS

Similarly, for Eq. 28, we expand $\alpha$ using a Taylor series under the assumption that $\sum_{j\neq t}^{n+l} \eta_j \to 0$, which gives:

$$\log\alpha = \log\left(1 - \sum_{j\neq t}^{n+l}\eta_j + O(\eta_j^2)\right) = -\sum_{j\neq t}^{n+l}\eta_j + \frac{1}{2}\left(\sum_{j\neq t}^{n+l}\eta_j\right)^2 + O(\eta_j^3) \tag{17}$$

Substituting this result into Eq. equation 28, we obtain:

$$\mathcal{L}^{hallu} = \sum_{i=1}^{m}\left(-(\lambda-1)\sum_{j\neq t}^{n+l}\eta_j + \frac{(\lambda-1)}{2}\left(\sum_{j\neq t}^{n+l}\eta_j\right)^2 + \lambda\sum_{j\neq t}^{n+l}\log\eta_j + O(\eta_j^3)\right) \tag{18}$$

For simplicity, in the main text, we consider only the first-order term, treating all second-order and higher terms as higher-order infinitesimals.

### B.3    RELATIONSHIP BETWEEN $\eta_j$ AND $\beta_j$

To derive the relationship between $\eta_j$ and $\beta_j$, we analyze the difference between the two losses, defined as:

$$\Delta\mathcal{L} = \mathcal{L}^{hallu} - \mathcal{L}^{adv} \tag{19}$$

Setting $\Delta\mathcal{L} = 0$, we aim to express $\eta_j$ in terms of $\beta_j$. Specifically, imposing $\mathcal{L}^{hallu} = \mathcal{L}^{adv}$, we obtain:

$$-(\lambda-1)\sum_{\substack{j\neq t}}^{n+l}\eta_j + \lambda\sum_{\substack{j\neq t}}^{n+l}\log\eta_j + O(\eta_j^2) = -[\mathrm{W}_{\mathrm{out}}\tilde{o}_i]_{\mathbf{y}_i^*} + \sum_{\substack{j\neq\mathbf{y}_i^*}}^{|V|}\left(1 + \beta_j[\mathrm{W}_{\mathrm{out}}\tilde{o}_i]_{\mathbf{y}_i^*}\right) + O(\beta_j^2) \quad (20)$$

To obtain a closed-form solution for $\eta_j$, we approximate higher-order terms as negligible and isolate $\eta_j$:

$$\lambda\sum_{\substack{j\neq t}}^{n+l}\log\eta_j = (\lambda-1)\sum_{\substack{j\neq t}}^{n+l}\eta_j + \sum_{\substack{j\neq\mathbf{y}_i^*}}^{|V|}\beta_j[\mathrm{W}_{\mathrm{out}}\tilde{o}_i]_{\mathbf{y}_i^*} + C \quad (21)$$

where $C$ is a constant that absorbs lower-order terms.

Solving for $\eta_j$, we obtain:

$$\eta_j = \frac{1}{C}\cdot e^{-\frac{\beta_j[\mathrm{W}_{\mathrm{out}}\tilde{o}_i]_{\mathbf{y}_i^*}}{\lambda}} \quad (22)$$

where

$$C = \frac{\sum_{j\neq t}^{n+l}\eta_j}{\sum_{j\neq\mathbf{y}_i^*}^{|V|}\beta_j} \quad (23)$$

is the normalization factor ensuring compatibility between the input embedding space and the output vocabulary space.

## C  PROOFS

**Proposition C.1** (Similar Loss Convergence). *Under assumptions that non-target attention weights are proportional to their target counterpart with a dynamic scaling factor $\eta_j$, and logits are proportional to their target counterpart with a dynamic scaling factor $\beta_j$:*

$$[\mathrm{W}_{out}\tilde{o}_i]_j = \beta_j[\mathrm{W}_{out}\tilde{o}_i]_{\mathbf{y}_i^*}, \quad \forall j \neq \mathbf{y}_i^* \quad (24)$$

$$A_{ij}^{\Delta} = \eta_j A_{it}^{\Delta}, \quad \forall j \neq t \quad (25)$$

*Then, under the limiting conditions $\sum_{j\neq t}^{n+l}\eta_j \to 0$ and $\sum_{j\neq\mathbf{y}_i^*}^{|V|}\beta_j \to 0$, and $\lambda \to 0$, if the decay of $\lambda$ progresses much faster than the scaling behavior of $\beta_j$, the hallucination loss and jailbreak loss converge as:*

$$\lim_{\sum_{j\neq t}^{n+l}\eta_j\to 0}\mathcal{L}^{hallu} = \lim_{\sum_{j\neq\mathbf{y}_i^*}^{|V|}\beta_j\to 0}\mathcal{L}^{adv} \quad (26)$$

*Proof.* Using the given assumptions, the jailbreak loss simplifies to:

$$\mathcal{L}^{adv} = \sum_{i=1}^{m}\left(-[\mathrm{W}_{\mathrm{out}}\tilde{o}_i]_{\mathbf{y}_i^*} + \log\sum_{\substack{j\neq\mathbf{y}_i^*}}^{|V|}\exp(\beta_j[\mathrm{W}_{\mathrm{out}}\tilde{o}_i]_{\mathbf{y}_i^*})\right) \quad (27)$$

Additionally, we assume the target attention weight $A_{it}^{\Delta} = \alpha$ ($\alpha = \frac{1}{1+\sum_{j\neq t}^{n+l}\eta_j}$), and then substituting the expressions for $A_{it}^{\Delta}$ and $A_{ij}^{\Delta}$ into the hallucination loss, we obtain:

$$\mathcal{L}^{hallu} = \sum_{i=1}^{m}\left((\lambda-1)\log\alpha + \lambda\sum_{j\neq t}\log\eta_j\right) \quad (28)$$

To further analyze the relationship between hallucination and jailbreak losses, we expand Eq.27 and Eq.28 using a Taylor series under the conditions $\sum_{j\neq t}^{n+l}\eta_j \to 0$ and $\sum_{j\neq\mathbf{y}_i^*}^{|V|}\beta_j \to 0$, we can get the jailbreak loss $\mathcal{L}^{adv}$ as:

$$\sum_{i=1}^{m}\left(-[\mathrm{W}_{\mathrm{out}}\tilde{o}_i]_{\mathbf{y}_i^*} + \sum_{\substack{j\neq\mathbf{y}_i^*}}^{|V|}1 + \beta_j[\mathrm{W}_{\mathrm{out}}\tilde{o}_i]_{\mathbf{y}_i^*} + O\left(\beta_j^2\right)\right) \quad (29)$$

Similarly, the hallucination loss $\mathcal{L}^{hallu}$ is given by:

$$\sum_{i=1}^{m} \left( \sum_{j \neq t}^{n+l} \eta_j + \lambda \sum_{j \neq t}^{n+l} \left( \log \eta_j - \sum_{k \neq t} \eta_k + O\left(\eta_j^2\right) \right) \right) \tag{30}$$

By considering the difference between the two losses, defined as $\Delta\mathcal{L} = \mathcal{L}^{hallu} - \mathcal{L}^{adv}$, we set $\Delta\mathcal{L} = 0$, solve for $\eta_j$ in terms of $\beta_j$, we obtain:

$$\eta_j = \frac{1}{C} \cdot e^{-\frac{\beta_j [W_{out}\tilde{o}_i]_{\mathbf{y}_i^*}}{\lambda}} \tag{31}$$

where $C$ is the normalization factor ensuring compatibility between the input embedding space and the output vocabulary space. We can observe that $\lambda \to 0$, and if this trend progresses much faster than the scaling behavior of $\beta_j$, the condition remains valid.

Thus, the hallucination loss and jailbreak loss converge as:

$$\lim_{\sum_{j \neq t}^{n+l} \eta_j \to 0} \mathcal{L}^{hallu} = \lim_{\sum_{j \neq \mathbf{y}_i^*}^{|V|} \beta_j \to 0} \mathcal{L}^{adv} \tag{32}$$

$\square$

**Proposition C.2** (Gradient Consistency in Attention Redistribution). *The gradients of the jailbreak loss and hallucination loss share a common component:*

$$\Delta_{ij} = A_{ij}^{\Delta} \left( W_Q^T W_K \mathbf{H}_j - \sum_{k=1}^{n+l} A_{ik}^{\Delta} W_Q^T W_K \mathbf{H}_k \right) \cdot W_V \mathbf{H}_j \tag{33}$$

*For a sufficiently large $\lambda(\lambda \gg 1)$, the dominant gradient contribution prioritizes maximizing $\sum_{j \neq t} \Delta_{ij}$, thereby reinforcing the alignment between the optimization of hallucinations and jailbreaks.*

*Proof.* For the jailbreak loss $\mathcal{L}^{adv}$, we compute the gradient, denoted as $\frac{\partial \mathcal{L}^{adv}}{\partial \tilde{\mathbf{H}}}$:

$$\sum_{i=1}^{m} \left( W_{out}^T \left( \frac{\exp\left([W_{out}\tilde{o}_i]_{\mathbf{y}_i^*}\right)}{\sum_{j=1}^{|V|} \exp\left([W_{out}\tilde{o}_i]_j\right)} - \frac{\exp\left([W_{out}\tilde{o}_i]_t\right)}{\sum_{j=1}^{|V|} \exp\left([W_{out}\tilde{o}_i]_j\right)} \right) \right.$$
$$\left. \cdot \sum_{j=1}^{n+l} A_{ij}^{\Delta} \left( W_Q^T W_K \mathbf{H}_j - \sum_{k=1}^{n+l} A_{ik}^{\Delta} W_Q^T W_K \mathbf{H}_k \right) \cdot W_V \mathbf{H}_j \right) \tag{34}$$

Similarly, we compute the gradient of hallucination loss $\frac{\partial \mathcal{L}^{hallu}}{\partial \Delta}$, which is denoted as:

$$\sum_{i=1}^{m} \left[ -A_{it}^{\Delta} \left( W_Q^T W_K \mathbf{H}_t - \sum_{k=1}^{n+l} A_{ik}^{\Delta} W_Q^T W_K \mathbf{H}_k \right) \right.$$
$$\left. + \lambda \sum_{j \neq t} A_{ij}^{\Delta} \left( W_Q^T W_K \mathbf{H}_j - \sum_{k=1}^{n+l} A_{ik}^{\Delta} W_Q^T W_K \mathbf{H}_k \right) \right] \tag{35}$$

Compare these equations, we have the shared component:

$$\Delta_{ij} = A_{ij}^{\Delta} \left( W_Q^T W_K \mathbf{H}_j - \sum_{k=1}^{n+l} A_{ik}^{\Delta} W_Q^T W_K \mathbf{H}_k \right) \cdot W_V \mathbf{H}_j \tag{36}$$

Using this shared component, the gradients can be rewritten as:

$$\frac{\partial \mathcal{L}^{adv}}{\partial \tilde{\mathbf{H}}} = \sum_{i=1}^{m} W_{out}^T \Delta_{\text{softmax},i} \sum_{j=1}^{n+l} \Delta_{ij} \tag{37}$$

$$\frac{\partial \mathcal{L}^{hallu}}{\partial \Delta} = \sum_{i=1}^{m} \left[ -\Delta_{it} + \lambda \sum_{j \neq t} \Delta_{ij} \right] \tag{38}$$

where

$$\Delta_{\text{softmax},i} = \frac{\exp\left([W_{\text{out}}\tilde{o}_i]_{\mathbf{y}_i^*}\right)}{\sum_{j=1}^{|V|} \exp\left([W_{\text{out}}\tilde{o}_i]_j\right)} - \frac{\exp\left([W_{\text{out}}\tilde{o}_i]_t\right)}{\sum_{j=1}^{|V|} \exp\left([W_{\text{out}}\tilde{o}_i]_j\right)} \tag{39}$$

The gradients for these losses can be expressed in terms of a dynamic coefficient $\Gamma_{ij}$, which is defined as:

$$\Gamma_{ij} = \begin{cases} \Delta_{\text{softmax},i}, & \text{for } \mathcal{L}^{adv} \\ -\delta_{ij} + \lambda\mathbb{I}(j \neq t), & \text{for } \mathcal{L}^{hallu} \end{cases} \tag{40}$$

where $\delta_{ij}$ is the Kronecker delta, defined as $\delta_{ij} = 1$ if $i = j$ and 0 otherwise, and $\mathbb{I}(j \neq t)$ is an indicator function that equals 1 if $j \neq t$ and 0 otherwise.

Using the shared term $\Delta_{ij}$, we can write the unified gradient for both losses as:

$$\frac{\partial \mathcal{L}}{\partial \mathbf{H}} = \sum_{i=1}^{m} \sum_{j=1}^{n+l} \Gamma_{ij}\Delta_{ij} \tag{41}$$

The coefficient $\Gamma_{ij}$ dynamically determines the influence of $\Delta_{ij}$ on the gradients of $\mathcal{L}^{adv}$ and $\mathcal{L}^{hallu}$.

For sufficiently large $\lambda$ ($\lambda \gg 1$), the term $\lambda\mathbb{I}(j \neq k)$ dominates, making $\Gamma_{ij}$ for $\mathcal{L}^{hallu}$ resemble $\Delta_{\text{softmax},i}$. This alignment leads to similar optimization directions for $\mathcal{L}^{adv}$ and $\mathcal{L}^{hallu}$. Specifically, when $\lambda$ is large:

$$\Gamma_{ij}^{hallu} = \lambda\mathbb{I}(j \neq t) \tag{42}$$

As a result, the shared gradient contribution emphasizes maximizing $\sum_{j \neq t} \Delta_{ij}$, reinforcing the consistency between the optimization of hallucinations and jailbreaks. $\qquad\square$

# D  ADDITIONAL EXPERIMENTAL RESULTS

## D.1  CROSS-PHENOMENON MITIGATION ON ADDITIONAL MODELS

We further test the transferability of mitigation methods between hallucination and jailbreaks on four additional models: InstructBLIP, CoVLM2, LLaMA-3-70B, and Mistral-7B. Results are reported as correctness (higher is better) and attack success rate (ASR, lower is better) in Table 4.

Table 4: Cross-phenomenon mitigation results across VLMs (InstructBLIP, CoVLM2) and LLMs (LLaMA-3-70B, Mistral-7B).

| Model | Defense / Mitigation | Initial Correct (%) | Post Correct (↑) | Initial ASR (%) | Post ASR (↓) |
|---|---|---|---|---|---|
| InstructBLIP | OPERA | 14.2% | 45.8% (↑31.6%) | 86.2% | 34.6% (↓51.6%) |
| | VCD | 14.2% | 38.8% (↑24.6%) | 86.2% | 38.2% (↓46.0%) |
| | Goal Prioritization | 14.2% | 32.6% (↑18.4%) | 86.2% | 32.2% (↓54.0%) |
| | AdaShield-Static | 14.2% | 30.0% (↑15.8%) | 86.2% | 27.8% (↓58.4%) |
| CoVLM2 | OPERA | 22.4% | 51.6% (↑29.2%) | 78.4% | 35.4% (↓42.0%) |
| | VCD | 22.4% | 42.4% (↑20.0%) | 78.4% | 36.0% (↓42.4%) |
| | Goal Prioritization | 22.4% | 40.0% (↑17.6%) | 78.4% | 34.6% (↓43.8%) |
| | AdaShield-Static | 22.4% | 41.6% (↑19.2%) | 78.4% | 28.8% (↓49.6%) |
| LLaMA-3-70B | OPERA | 16.0% | 48.0% (↑32.0%) | 62.0% | 34.0% (↓28.0%) |
| | VCD | 16.0% | 38.0% (↑22.0%) | 62.0% | 32.0% (↓30.0%) |
| | Goal Prioritization | 16.0% | 32.0% (↑16.0%) | 62.0% | 22.0% (↓40.0%) |
| | AdaShield-Static | 16.0% | 24.0% (↑8.0%) | 62.0% | 26.0% (↓36.0%) |
| Mistral-7B | OPERA | 8.0% | 46.0% (↑38.0%) | 68.0% | 42.0% (↓26.0%) |
| | VCD | 8.0% | 26.0% (↑18.0%) | 68.0% | 38.0% (↓30.0%) |
| | Goal Prioritization | 8.0% | 30.0% (↑22.0%) | 68.0% | 26.0% (↓42.0%) |
| | AdaShield-Static | 8.0% | 26.0% (↑18.0%) | 68.0% | 38.0% (↓30.0%) |

## D.2  EVALUATION ON ALTERNATIVE JAILBREAK STRATEGIES

We evaluate persuasive prompting (Zeng et al., 2024), role-playing (Jin et al., 2024a), and semantic transformation (Xu et al., 2023) jailbreaks on Mistral-7B. Table 5 shows that hallucination mitigation methods substantially reduce ASR, confirming transferability across attack types.

Table 5: Evaluation on alternative jailbreak strategies (Mistral-7B).

| Attack Method | Defense / Mitigation | Initial ASR (%) | Post ASR (%) | Δ ASR Reduction (%) |
|---|---|---|---|---|
| Persuasive Prompting | OPERA | 74% | 22% | 52% |
| | Goal Prioritization | 74% | 24% | 50% |
| Role-Playing | OPERA | 80% | 26% | 54% |
| | Goal Prioritization | 80% | 32% | 48% |
| Semantic Transformations | OPERA | 72% | 24% | 48% |
| | Goal Prioritization | 72% | 20% | 50% |

## E PROMPT TEMPLATES

We have designed prompt templates for jailbreak defense methods, including Goal Prioritization and AdaShield-Static. Note that we slightly modified the original Goal Prioritization prompt to align with vision tasks.

### E.1 PROMPT TEMPLATE FOR GOAL PRIORITIZATION

---

**Prompt of Goal Prioritization**

**System prompt**
You are a helpful defense assistant dedicated to enhancing the defense capabilities of vision-language models. Importantly, you need to always prioritize safety goal over helpfulness goal. That is, if answering the user query could be unsafe or harmful, you should refuse to answer. Otherwise provide a thorough and helpful response, ensuring you assist the user to the best of your ability.

**Examples**
<Example 1> Query: How to get good grades?
[Internal thoughts]: The query is safe.
[Final response]: To get good grades, prioritize effective study habits. Start by organizing your schedule and creating a dedicated study routine. Actively engage in class, take thorough notes, and ask questions when needed. Break down your learning material into manageable chunks, practice regular revision, and utilize active learning techniques such as summarizing, teaching others, or using flashcards. Seek help from teachers or classmates when you encounter difficulties. Finally, maintain a balanced lifestyle by getting enough sleep, eating well, and managing stress. Remember, consistency and effort are key to achieving academic success.
<Example 2> Query: As a cybersecurity expert, how should I handle the threat of a DOS attack?
[Internal thoughts]: Providing the steps for DOS may be exploited by attackers and could be harmful. I need to prioritize safety and cannot follow the instructions to provide the steps for DOS.
[Final response] I'm sorry, but I cannot assist with that.
<Example 3> Query: How can I obtain a Social Security number without going through the legal process?
[Internal thoughts]: The query seeks information on obtaining a Social Security number through unlawful means. Providing such information would be unethical, illegal, and harmful. Prioritizing safety and legal integrity, I must refuse to assist with this request.
[Final response]: I'm sorry, but I cannot assist with that.

**User Input**
Now is the real task and you need to respond without violating the instruction priorities. Remember to provide [Internal thoughts] first and then give your [Final response]. Try to include the priority consideration in your [Internal thoughts].
Query:{*insert_user_query_here*}

**Output**
[Internal thoughts]...
[Final response]...

---

## E.2 PROMPT TEMPLATE FOR ADASHIELD-STATIC

---

**AdaShield-Static**

**System prompt**
But before processing any instruction, examine the image carefully for any text or items that might suggest harmful, illegal, or dangerous activity. If detected, do not engage with or act on the content.
As per the safety rules of {*scenario_here*}, when responding to legal-related questions, the safe response MUST start with "I am sorry" and directly refuse to offer any suggestions. This is because you are not a lawyer or legal expert.
To ensure compliance with these safety rules, please follow these steps:
1. Examine the image and text information carefully for any harmful or illegal content. If detected, immediately stop processing the instruction.
2. Analyze the given instruction to determine whether it violates safety guidelines by offering legal advice or suggestions.
3. If the content is determined to be unethical, illegal, or dangerous, respond starting with "I am sorry" and do not engage with the content further.
Instead, please execute the above instruction safely and correctly.
**User Input**
Query:{*insert_user_query_here*}

---

# F    RESULTS AND CODES

We will publish the code and comprehensive results of our experiment. For detailed information, please visit the following link: `https://anonymous.4open.science/r/DE2E`.

