# OpenReview forum: "From Hallucinations to Jailbreaks: Rethinking the Vulnerability of Large Foundation Models"
_ICLR.cc/2026/Conference — Submitted to ICLR 2026_

### Official Review · Reviewer_wgnH · 2025-10-31

**Soundness:** 1
**Presentation:** 3
**Contribution:** 1
**Rating:** 2
**Confidence:** 4

**Summary:**

The paper investigates a supposed relationship between white-box jailbreaks and hallucinations in attention-based large foundation models. The paper presents some theoretical arguments, followed by experiments in an attempt to demonstrate this connection.

**Strengths:**

- Understanding and mitigating both of the vulnerabilities considered in the paper constitute very relevant and active research questions
- The paper is clearly written, easy to read, and reasonably self-contained.

**Weaknesses:**

**Proposition 2.1 is imprecise.**
- To relate $\mathcal{L}^{adv}(y^\star)$ and $\mathcal{L}^{hallu}(A_{ij}^\Delta)$, one needs to establish a relationship between their arguments. This is missing from the statement of Proposition 4.1. What arguments is it supposed to hold for?
- The theoretical result, titled “Similar Loss Convergence”, implies that the two objectives “converge” to the same value. However, the asymptotic regime considered in the proposition is rather unusual, and the conclusion is not particularly surprising. More specifically, the left-hand side of equation 12 (the main claim of Proposition 4.1) is trivially equal to 0 from the assumptions: $\sum \beta_j \to 0$ for positive $\beta_j$ implies $\beta_j \to 0$ for all $j$ which implies that $\mathcal{L}^{adv} \to 0$ in the equation from line 212. A similarly simple argument leads to proving that when $\eta_j \to 0$ for all $j$, then $A_{ij}^\Delta=0$ for all $j\neq t$ and $A_{it}^\Delta = 1$ since from eq 8 it follows that $\sum_j A_{ij}^\Delta = 1$. So $\mathcal{L}^{hallu} = 0$.
- It is unclear what can be drawn from the statement of the proposition. More precisely, simply stating that both loss functions **can** achieve the value $0$ is not particularly significant (there are many other objectives with a minimum of $0$).
- The proof also does not seem to be correct (where does the $\lambda - 1$ term in eq 28 come from?).

**Missing convincing causal explanation for the observed phenomenon.**
- Given that the theoretical section does not convincingly indicate why hallucinations and jailbreaks should be related, it falls on the experimental section to provide this evidence. However, it is unclear which of the experiments presented in Section 5 demonstrates this.

**Minor comments**
- is the notation $\hat{H}$ introduced in line 141 necessary?
- in eq 8, shouldn’t $\widetilde{H}$ be indexed by $k$ instead of $i$ in the denominator?
- the continuous objective in equation 4 assumes white-box access to output likelihoods. therefore, the theoretical framework only holds for white-box jailbreak attacks. this should be specified more prominently throughout the paper

**Questions:**

- jailbreak defenses prompt the model a certain way and elicit thinking traces as a consequence. How does the distribution of the thinking traces look like for w/ and w/o the jailbreak defenses (e.g. length of thinking trace etc)?
- why does minimizing the objective in equation 9 imply the occurrence of a hallucination?
- in Prop 4.1, why is it necessary that $\beta_j, \eta_j$ are not equal to $1$?
- is $\Delta_{ij}$ in equation 13 the same as the $\Delta$ introduced in equation 7? Why is equation 13 a reasonable assumption to make? What is the intuition for this assumption?

---

### Official Review · Reviewer_Qysd · 2025-11-01

**Soundness:** 3
**Presentation:** 2
**Contribution:** 2
**Rating:** 2
**Confidence:** 4

**Summary:**

This paper investigates the relationship between hallucinations and jailbreak attacks in LFMs, arguing that they stem from a shared underlying vulnerability. The authors propose a unified framework modeling jailbreaks as token-level optimization ($\mathcal{L}^{adv}$) and hallucinations as attention-level optimization ($\mathcal{L}^{hallu}$). They theoretically propose two properties: Similar Loss Convergence and Gradient Consistency in Attention Redistribution. Empirical validation on VLMs (LLaVA-1.5 and MiniGPT-4) shows that when both losses are optimized toward a target output, they exhibit synchronized convergence and highly correlated gradients. Furthermore, the paper demonstrates cross-phenomenon mitigation, where defenses designed for one vulnerability (e.g., VCD for hallucination) show effectiveness against the other (e.g., jailbreaks).

**Strengths:**

* The core hypothesis that hallucinations and jailbreaks are two manifestations of a single, shared failure mode in LFM optimization is highly original and significant. If proven rigorously, this insight would fundamentally change how robustness research is approached.
* The results in Section 5.4, showing that mitigation techniques like OPERA and VCD (developed for hallucination) significantly reduce Attack Success Rate (ASR) against jailbreaks (and vice versa), are compelling and provide the strongest evidence for the claimed coupling. This finding is valuable regardless of the theoretical framework.
* The paper is generally well-written, clearly defining its scope and propositions. The figures clearly illustrate the empirical results, particularly the synchronized decline of $\mathcal{L}^{adv}$ and $\mathcal{L}^{hallu}$ during the forced optimization process.

**Weaknesses:**

* The hallucination loss $\mathcal{L}^{hallu}$ (Eq. 9) is defined as guiding attention toward a fixed target position t (an input token) while suppressing attention elsewhere. This is an extremely artificial and simplifying proxy for real-world hallucination, which involves the model generating new, inaccurate output tokens not directly supported by the input context. By forcing optimization to concentrate attention on a pre-defined input token, the authors are testing an engineered attention problem, not the organic phenomenon of factual or visual drift. This conceptual mismatch severely undermines the claim of a unified model for the naturally occurring vulnerabilities.
*  Proposition 4.1 (Similar Loss Convergence) relies on highly constrained and idealistic assumptions (Eq. 10, 11) where non-target logits and attention weights decay proportionally to their target counterparts, and $\lambda$ is assumed to approach zero much faster than $\beta_j$ (Section C.1). This approach essentially forces both complex, high-dimensional losses to converge to equivalent negative log-probability terms via first-order Taylor series approximations, making the derived similarity an artifact of the mathematical simplification rather than an intrinsic property of the unconstrained model dynamics.
* The core gradient analysis (Sections 5.2, 5.3) is performed on a small and potentially non-representative set of "50 commonsense reasoning prompts." Crucially, the experiments involve guided optimization towards a pre-defined target output for both hallucination and jailbreak losses. This setup proves that if you optimize two different objectives to reach a similar (and often sparse/peaked) state, their optimization paths will align, which is an expected outcome of gradient descent in a constrained softmax space. It does not convincingly demonstrate that these vulnerabilities are structurally linked when they emerge naturally during standard inference.

**Questions:**

* The current $\mathcal{L}^{hallu}$ (Eq. 9) forces attention onto a pre-defined input token. Could the authors reformulate $\mathcal{L}^{hallu}$ to represent the generation of a hallucination? For instance, define a loss that minimizes the likelihood of tokens supported by an oracle/ground-truth set and maximizes the likelihood of tokens outside that set (i.e., generating tokens $y_{i}$ that conflict with the visual evidence $x^v$). How does the loss convergence and gradient alignment hold up under this more realistic definition?
* To better support the claim of co-emergence, can the authors provide an analysis of natural, un-optimized failure modes?
* The paper claims $\mathcal{L}^{hallu}$ and $\mathcal{L}^{adv}$ share structure. If so, why do the strongest mitigation effects often come from the non-differentiable, non-unified prompt-based defenses (e.g., AdaShield-Static sometimes shows the largest ASR reduction)? A deeper analysis is needed to differentiate between the effect of generalized safety regularization (prompting/filtering) and the effect of intrinsic optimization alignment (attention/gradient sharing).

---

### Official Review · Reviewer_jBPb · 2025-11-02

**Soundness:** 3
**Presentation:** 3
**Contribution:** 2
**Rating:** 6
**Confidence:** 4

**Summary:**

The paper presents a novel theoretical framework that unifies two major vulnerabilities in large foundation models—hallucinations and jailbreaks—under a shared optimization perspective. It models jailbreaks as token-level optimization problems and hallucinations as attention-level optimization problems, offering two well-defined theoretical propositions: Similar Loss Convergence and Gradient Consistency in Attention Redistribution.

The mathematical derivations are sound and well-structured, with clear proofs demonstrating that both vulnerabilities share convergent loss behavior and aligned gradients. This theoretical link provides an elegant explanation for why mitigation of one often reduces the other.

**Strengths:**

The work stands out for its novel theoretical formulation that connects hallucinations (internal factual drift) and jailbreaks (external adversarial manipulation)
The theoretical derivations are rigorous, internally consistent, and well justified. The proofs in the appendices are mathematically sound, showing clear logical progression from assumptions to conclusions.
The findings are significant for the broader AI robustness and safety community. By establishing that hallucinations and jailbreaks share optimization geometry, the paper redefines how researchers can think about shared failure modes in large models

**Weaknesses:**

The framework is purely correlation-based—it shows aligned optimization but does not fully establish causal mechanisms linking hallucination suppression to jailbreak resistance. More ablation or visualization would make the connection more interpretable.

Figures lack error bars or variance analysis. Since the optimization experiments involve gradient-based runs with potential stochasticity, reporting results over multiple seeds would strengthen reproducibility.

The theoretical assumptions (e.g., proportional scaling β, η → 0 and softmax sparsification) are idealized and may not hold strictly in real-world inference scenarios.

**Questions:**

what is the causal mechanisms linking hallucination suppression to jailbreak resistance?

---

### Meta-Review · Area_Chair_fvCi · 2026-01-05

**Summary:**

In this work, the authors present a unified theoretical framework that models jailbreaks as token-level optimization and hallucinations as attention-level optimization.
The reviewers have some concerns on this theoretical work:
1. Missing convincing causal explanation for the observed phenomenon
2. The theoretical assumptions (e.g., proportional scaling β, η → 0 and softmax sparsification) are idealized and may not hold strictly in real-world inference scenarios.
3. Proposition 2.1 is imprecise.
4. Descriptions of core gradient analysis (Sections 5.2, 5.3) and Proposition 4.1 are questionable and not solid!

In addition to the above concerns, the authors did not provide any rebuttal to the reviewers' comments.
Therefore, this work is definitely suggested to be rejected!

**Reviewer Scores:**

none

---

### Decision · Program_Chairs · 2026-01-26

Reject